# Healthy Food Options at Dollar Discount Stores Are Equivalent in Quality and Lower in Price Compared to Grocery Stores: An Examination in Las Vegas, NV

**DOI:** 10.3390/ijerph15122773

**Published:** 2018-12-07

**Authors:** Courtney Coughenour, Timothy J. Bungum, M. Nikki Regalado

**Affiliations:** School of Community Health Sciences, University of Nevada, 4505 S. Maryland Parkway Box 3064, Las Vegas, NV 89154, USA; tim.bungum@unlv.edu (T.J.B.); Regalm2@unlv.nevada.edu (M.N.R.)

**Keywords:** food access, health disparity, food desert, community food environment, channel blurring, food affordability, Las Vegas, health equity

## Abstract

Food deserts indicate limited access to and affordability of healthy foods. One potential mediator is the availability of healthy food in non-traditional outlets such as dollar-discount stores, stores selling produce at the fixed $1 price. The purpose of this study was to compare availability, quality, price differences in ‘healthier’ versus ‘regular’ food choices, price per each food item, and summary score in dollar-discount stores to grocery stores in Las Vegas using the NEMS-S; a protocol consisting of three subscores—availability, quality, price of healthier versus regular food, and a summary score. A 25% sample of grocery stores (*n* = 40) and all dollar-discount stores (*n* = 14) were evaluated. *t*-tests showed that dollar-discount stores were less likely to price healthy options lower than their unhealthy alternatives (mean (M) = 1.0 vs. M = 2.5; *p* < 0.001) and had reduced availability (M = 20.50 vs. M = 23.80; *p* < 0.001) compared to grocery stores. The quality of produce did not differ (M = 5.93 vs. M = 6.00; *p* = 0.34). Price comparisons revealed that 84.2% of produce and 89.5% of other food items were significantly less expensive at the dollar-discount stores, with only two items being more expensive. While dollar-discount stores did have lower availability, they provided quality fresh and healthy foods which were usually less expensive. Findings indicate that dollar discount stores may be an existing community asset, and considering them as such may aid in efforts to strengthen the overall food system. Practitioners should consider dollar discount stores when assessing the community food environment and designing and implementing outreach programs, as they may bridge some disparities in access.

## 1. Introduction

Consuming a nutritious diet is a pillar of good health. Yet, many studies have documented that access to retail food outlets that offer affordable, nutritious foods is lacking in some communities [1,2,3]. In a literature review examining access to food outlets in developed countries, White concludes that the United States “is a more unequal society, where issues such as food retail access are genuinely worse for the poor and, in particular, African Americans”. Nationally, 11.5 million people now live in low-income areas that are more than one mile from a grocery store. Such areas with limited access and limited resources necessary to access healthy foods are termed “food deserts” [4]. Living in a food desert may make it harder to consume a healthy diet. While findings on the associations between food access and health are mixed, studies have reported that lacking access to affordable, nutritious food is linked to negative health behaviors and outcomes, including decreased consumption of fruits and vegetables [5,6] and high obesity rates [7,8,9]. The affordability of healthy foods has also been linked to negative health behaviors and outcomes [10,11]. 

Equitable access to affordable, healthy food options is thus important for improving and maintaining one’s diet and overall health. However, the most effective way to accomplish this is not inherently clear. While it may be beneficial to build grocery stores within these communities, it is not always possible for multiple reasons. Before locating in an area, grocers typically conduct a cost/benefit analysis that may suggest that economic and market factors do not support such a siting unless coupled with favorable policy or tax interventions. Bitler & Haider provide the following example: suppose the population residing in a food desert largely consists of low income families who, though they may desire nutritious food, are unable to afford it [12]. In this example, the for-profit business model would suggest that the benefits of locating a store in these areas would be profitable only if food assistance benefits, such as the Supplemental Nutrition Assistance Program (SNAP), are expanded, and/or tax incentives or government subsidizing of higher operating costs are available. 

Additional efforts have been made to increase healthy food access, including the use of farmer’s or mobile markets [13], and in-store marketing of healthy items [14]. Because it is the role of public health to ensure that all people have the opportunity to access healthy, fresh food, exploring options that may already exist within the community to provide such access could prove to be beneficial. 

One phenomenon that may be useful is the practice of channel blurring. Channel blurring occurs when retailers from one channel sell products typically associated with other channels [15]. For example, drug stores now sell various food products, toys and household goods, big box stores sell a large variety of products, including groceries, and some discount or dollar stores sell fresh produce [15]. A growing number of these discount stores offer fresh produce and other healthy food options that would traditionally be found only in grocery stores. These non-traditional stores may serve to bridge the gap in providing access to healthy and affordable foods to communities who may otherwise lack access.

To determine if dollar discount stores are filling this gap, the current study used the Nutrition Environment Measures Survey in Stores (NEMS-S) [16] to compare availability, quality, price differences in the ‘healthier’ versus ‘regular’ food choices, NEMS-S summary score, and price per each food item found at grocery stores to those found at dollar discount stores in the Las Vegas Metropolitan area.

## 2. Materials and Methods

### 2.1. Sample Selection

In the summer of 2015, all grocery stores within the Las Vegas metropolitan area were identified using Google maps searches, ground truthing or verifying in person, and cross-referencing all retailers’ websites to ensure that all locations were documented. We used a slightly modified version of Jiao and colleagues supermarket definition: stores that are run by national or regional chains, or are comparable in size and scope, and also sell a broad selection of foods, such as canned and frozen foods; fresh fruit and vegetables; and fresh and prepared meats, fish, seafood, and poultry, regardless of the number of venues [17]. Next, 25% of all retail grocery outlets were systematically selected based on geographical distribution across the Las Vegas Metropolitan area, resulting in a sample of 40 grocery stores. Attempts were made to select an equal number of stores that represented each income quintile as defined by the 2014 American Community Survey five-year estimates of median household income for the census tract that each store was located within. See Figure 1 for a map of stores with corresponding income quintile. 

One chain of discount stores that regularly stock a produce section were located using the same methods that were employed for identifying grocery stores. This resulted in a sample of 14 dollar discount stores. See Figure 1 for a map of all discount stores with corresponding income quintile. 

### 2.2. NEMS-S Auditing Tool

All discount and grocery stores were evaluated using the NEMS-S, a valid and reliable tool [16] that is widely used [16,18,19,20,21]. NEMS-S measures retail food environments using three subscales, availability, quality, and price of healthier versus regular food, as well as a summary score. Following the NEMS-S protocol, 11 categories were evaluated for availability, earning higher points if there was a larger variety and healthier options (for example, two points are earned if whole grain bread is available, and an additional point is earned if there are greater than two varieties of whole grain bread). Total possible availability scores range from 0 to 30 points. The quality score is based on the acceptability of fruits and vegetables, with more points being earned based on the proportion of acceptable produce items. Acceptable is defined by the majority of the produce being in “peak condition, top quality, good color, fresh, firm and clean” [16]. Total possible quality scores range from 0 to 6 points. The price metric on the NEMS-S tool measures the price differences in the ‘healthier’ versus ‘regular’ options (i.e., price differences in baked versus regular potato chips), awarding more points if the healthy alternative is priced lower than the regular item and subtracting points if it is priced higher. Total possible price of healthier versus regular food scores range from −9 to 18 points. A NEMS-S summary score is determined by adding the points earned for availability, quality, and price of healthier versus regular food. The maximum NEMS-S summary score is 54 points.

We were also interested in comparing sale prices of each food item at grocery versus discount stores. Because produce was sometimes priced per pound, and sometimes priced per piece, all produce items were recalculated to price per piece using the USDA National Nutrient Database [22]. For example, one medium banana is estimated to weigh 0.26 pounds, thus there are 3.85 bananas per pound. If priced at $0.59 per pound, each banana was estimated to cost $0.15. Similarly, other perishable and non-perishable items were of differing sizes, so items were recalculated to price per ounce. For example, a 16-ounce loaf of whole wheat bread may be priced at $0.99, thus cost about $0.06 per ounce. 

### 2.3. Food Environment Audits

Audits of the grocery and dollar discount stores were conducted by two student researchers who had completed the free online training for the NEMS-S tool offered through the University of Pennsylvania. To ensure reliability, these researchers simultaneously and independently audited a grocery store and a dollar discount store. They then compared their responses to each item on the tool and resolved any discrepancies by talking through their reasoning. Once their audits were consistent, they separately continued auditing the remaining grocery and dollar discount stores. Researchers entered the store and completed the audit forms using pen, paper, and clipboards. Each audit took the researchers about 40 min to complete. On a few occasions, store personnel asked researchers what they were doing. The reply that they were “assessing the food environment” satisfied store personnel. All audits were completed without issue or confrontation.

### 2.4. Statistical Analysis

Independent sample *t*-tests were conducted using SPSS24 (IBM Corp. Armonk, NY, USA:) in order to compare the mean of the scores for availability, price of ‘healthier’ versus ‘regular’ food choice, quality, NEMS-S summary score, and sale price per each food item included in the NEMS-S tool at each of the two store types. Alpha was set at 0.05 to determine statistical significance.

## 3. Results

### 3.1. Availability

The *t*-test results showed that grocery stores had a significantly higher NEMS-S availability score (M = 23.80, SD = 2.54) compared to dollar discount stores (M = 20.50, SD = 1.29; t(52) = −6.24; *p* < 0.001). This indicates that grocery stores had a greater variety of food options compared to dollar discount stores. Of note, no dollar discount stores carried pears and low fat ground beef, and 57% did not carry regular ground beef. 

### 3.2. Quality

There was not a statistically significant difference between the quality of fresh fruits and vegetables at grocery stores (M = 6.00, SD = 0.0) compared to dollar discount stores (M = 5.93, SD = 0.27; t(13) = −1.00; *p* = 0.34). A lack of significant difference in the two mean scores indicated that the examined food at the dollar discount stores was equitable in quality to that at the grocery stores. 

### 3.3. Price of ‘Healthier’ versus ‘Regular’

Our results showed that grocery stores were significantly more likely to price healthy alternatives at a lower price (M = 2.55, SD = 2.48) compared to dollar discount stores (M = 1.0, SD = 0; t(39) = −3.95; *p* < 0.001). 

### 3.4. NEMS-S Summary Score

The results of the *t*-test showed that grocery stores had a significantly higher summary score (M = 32.35, SD = 3.70) compared to dollar discount stores (M = 27.43, SD = 1.28; t(52) = −7.25; *p* < 0.001).

### 3.5. Sale Price Per Each Food Item

The results of *t*-tests showed that 84.2% of produce and 89.5% of non-produce items were significantly less expensive at dollar discount stores. There was not a significant price difference between the grocery and dollar discount stores for bananas, watermelons, cucumbers, regular ground beef, low-sugar cereal, and regular chips. The only items that were more expensive at dollar discount stores were whole wheat and white bread. See Table 1 for full results of all food items.

## 4. Discussion

The most interesting findings of this study are that the quality of produce items did not differ between the grocery and dollar discount stores, and that most items were less expensive at the dollar discount stores. These findings are important for public health, as our study indicates that channel blurring at the dollar discount stores results in access to healthy, quality produce and affordable food options. Because cost, quality, and accessibility are established barriers to healthy eating [11,23,24,25], dollar discount stores can serve as community assets that increases access to quality, affordable food. Additionally, the dollar discount stores in our sample all accepted Supplemental Nutrition Assistance Program (SNAP) funds, further increasing the ability of those most needing to manage and stretch their food budgets. Public health researchers and practitioners should view these retailers as a community asset and work to collaborate on health promotion and outreach efforts. 

Overall, grocery stores did have significantly higher availability and NEMS-S summary scores. This is to be expected because our definition for grocery stores limited our sample to chain stores, or those that were comparable in size, which are larger stores that have more shelf space than the dollar discount stores. This definition was purposeful, as larger stores are able to offer products at lower prices due to their buying power [26,27], and we were seeking a comparable sample of grocery stores. Thus, our sample of grocery stores was exclusive to larger stores that offered a great variety of foods. In addition to being, on average, smaller store fronts, dollar discount stores are known for selling non-food products, such as party supplies and trinkets, which would compete for space that might otherwise be filled with food products. While the NEMS-S summary scores were higher in grocery stores, they appear to be a less useful indicator of a store’s ability to provide healthy foods at an affordable price; the sub-scales and added variable of sale price per each food item seem to be more informative.

Given that our grocery store definition limited us to larger stores for their lower prices, our finding that nearly all items were still less expensive or not significantly different at the dollar discount store is important. This definition precluded the inclusion of any smaller neighborhood markets that, while they are likely to be an asset to a community’s food environment, have higher prices [26]. Including the smaller neighborhood markets would have driven the mean prices of food up, further highlighting the dollar discount stores as an affordable option.

Grocery stores were more likely to price the healthy food alternatives at lower prices. While this finding was not surprising given that the dollar discount store priced most food items uniformly at dollar discount, a lower sticker price on the healthy food at point of sale may have an effect on food choices, as previous research has shown that reducing the price of the desired healthy choice does influence sales [28]. For example, French found that “price reductions of 10%, 25% and 50% on lower fat snacks resulted in an increase in sales of 9%, 39% and 93%, respectively” [29]. However, our findings indicated that the overall price of nearly all items at the dollar discount store were lower or did not differ from the grocery store, so it is difficult to speculate if consumer behaviors would result in the purchase of the non-healthy or healthy alternative given equal pricing at point of sale. Overall research findings support price modification as a tool to influence targeted food purchases [28]. From a public health perspective, lower price points for healthy alternatives are likely to impact those who are often more vulnerable to chronic disease and chronic disease indicators, such as low income individuals and those with less education, as they are forced to be more price conscious than those with more expendable income [30,31]. In addition to price modification, instore marketing of inexpensive foods with high nutritional value have also been shown to influence purchasing behaviors in some food categories [14]. Instore marketing may be viewed by retailers as an easier, sustainable alternative to price modification. 

It should be noted that the dollar discount stores did not always carry the same brand name products that are found in grocery stores. While the products themselves are likely to be equivalent nutritionally [32,33], marketing research confirms that pricing and branding influence perceptions of both quality and intention to buy. Studies have shown that consumers recognize brand names and perceive them to be of higher quality [34,35,36,37], even when they are not [32,33]. Brand loyalty may also influence purchases and perceptions of products at the dollar discount versus grocery stores, as those who are loyal to a specific brand may be unwilling to deviate from their normal buying habits [34,38]. 

Consumers also use price as an indicator of quality, thus, the lower price at the dollar discount store may result in perceptions of lower quality [35,39]. Market factors such as these may result in dollar discount store consumers either purchasing fewer products or feeling as though they are receiving an inferior product. Overcoming these market factors is likely to prove difficult, as changing brand recognition and reputation requires large investments in advertising [40]. Public health interventions in communities with low access to chain grocery stores may work to overcome this misperception by informing the community that the quality of produce did not differ at either location and that significantly lower prices at dollar discount stores can result in overall cost savings. This is particularly valuable information for vulnerable populations such as recipients of SNAP and those with limited incomes and/or food budgets.

The current study also has implications for public health practice. In addition to considering dollar discount stores and other non-traditional food outlets when assessing community food environments, public health researchers and practitioners should consider collaboration and partnership opportunities. For example, outreach programs such as the “Double Up Food Bucks” program that matches the value of SNAP funds when those funds are spent on fresh produce, might consider the dollar discount stores and other discount markets when looking for partners. Public health workers may also consider working with such stores to offer point-of-choice nutrition information, or manipulate the price differences between the non-healthy and healthy alternative foods to influence purchases of the desired healthy choices.

While dollar discount stores, pricing all items at a fixed $1 price, may be more common in the US, findings are still pertinent to other countries. For example, non-fixed price discount stores, as well as full service grocery stores that sell non-brand name food items are likely to offer equally healthy foods at a lower price. Hence, compared with branded products, non-branded alternatives are nutritionally equivalent options for a better price, and should therefore be preferred options to improve dietary quality when the budget for food is low.

This study assessed the food environment of a sample of grocery stores and dollar discount stores in the Las Vegas metropolitan area, and is not without limitations. While most of the stores we assessed are regional or national chains, our findings may not be generalizable to other locations. Although research supports a relationship between food price, quality, and availability, because we only assessed the food environment, we cannot be certain how this influences the relationship with health behaviors and outcomes. Our study was carried out in the summer, a season when fresh fruits and vegetables are more plentiful due to the growing seasons. It is unknown how the price, quality, and availability of produce and other food items would be affected in non-summer months. 

## 5. Conclusions

Our findings have significant implications for public health. While dollar-discount stores did have lower availability compared to grocery stores, they provided quality fresh and healthy foods which were usually less expensive. Retail outlets such as the dollar discount stores can serve as a community asset, offering quality, affordable food options and enhancing accessibility. This is especially important for individuals that are most vulnerable for poor health outcomes, such as those with low income, low educational attainment, and SNAP recipients. It is important for practitioners to understand the local food environment completely in order to document all opportunities for healthy food options, and to consider these non-traditional food outlets when assessing community food environments, designing and implementing nutritional interventions, and providing information or nutritional advice to those who do not live near traditional grocery stores. Additionally, considering dollar discount stores in food improvement efforts would strengthen the overall food system, such as nutrition labeling efforts or incorporation of urban agriculture efforts to offer more locally produced foods. Seeing these retailers as a community asset and working with them can aid in health promotion and outreach efforts. Recognizing and utilizing community resources is especially important given that public health funding is often limited and inadequate for meeting needs [41]. 

## Figures and Tables

**Figure 1 ijerph-15-02773-f001:**
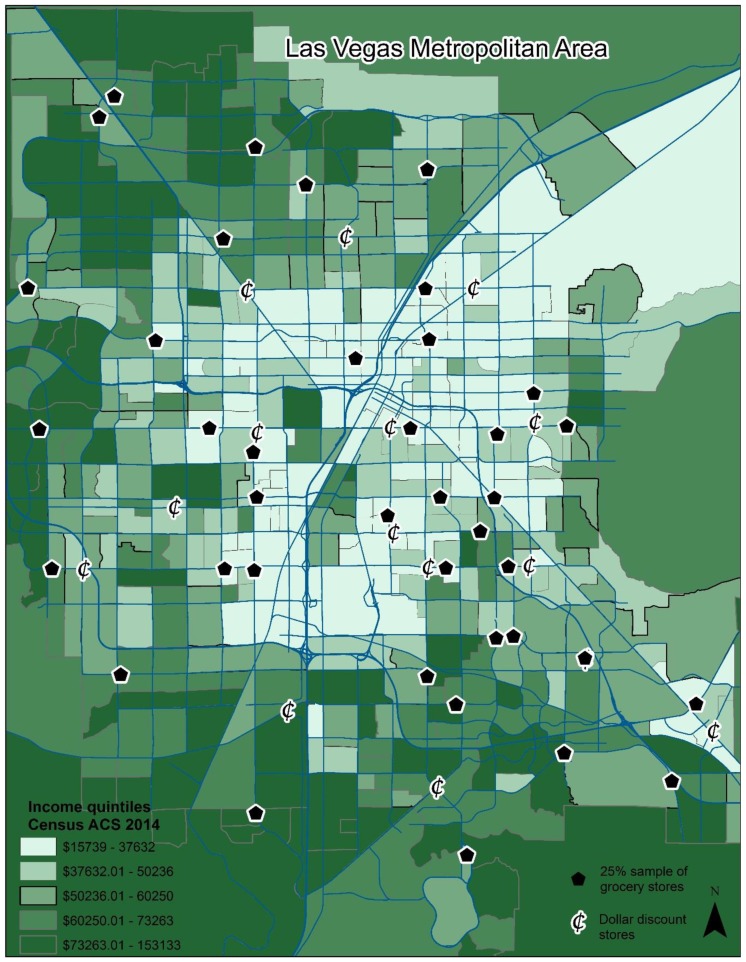
Map depicting the dollar discount stores audited (*n* = 14) and grocery stores audited (*n* = 40) representing a 25% sample of all grocery stores. Income quintiles based on Clark County, NV median household income per the 2014 U.S. Census, American Community Survey (ACS) five-year estimates.

**Table 1 ijerph-15-02773-t001:** *t*-test results for price of each produce and food item at grocery (*n* = 40) and dollar discount stores (*n* = 14) in Las Vegas Metropolitan area in summer 2015.

Food Type	Store Type								
	Grocery Store			Dollar Discount Store					
	M	SD	n	M	SD	n	95% CI	t	df
Banana	0.15	0.03	40	0.18	0.10	14	−0.085, 0.037	−0.85	13.8
Apple	0.62	0.25	40	0.36	0.27	8	0.061, 0.046	2.63 *	46.0
Orange	0.48	0.60	40	0.12	0.32	12	0.005, 0.704	2.04 *	50.0
Grapes	0.87	0.52	36	0.28	0.0	13	0.297, 0.882	4.05 ***	47.0
Cantaloupe	2.00	0.87	40	0.99	0.0	14	0.730, 1.29	4.31 ***	39.0
Peach	0.82	0.35	19	0.20	0.07	8	0.440, 0.793	7.25 ***	21.4
Strawberry	0.96	0.30	40	0.32	0.0	12	0.548, 0.740	13.56 ***	39.0
Honeydew Melon	3.15	0.85	39	1.19	0.46	12	1.580, 2.36	10.29 ***	35.0
Watermelon	3.76	0.95	40	1.88	2.80	10	−0.131, 3.905	2.10	9.5
Pear	0.67	0.27	40	NA		0			
Carrot	0.11	0.03	40	0.08	0.02	12	0.017, 0.054	3.85 ***	50.0
Tomato	0.47	0.21	40	0.34	0.15	14	0.007, 0.255	2.12 *	52.0
Sweet Pepper	0.95	0.52	40	0.50	0.0	11	0.287, 0.616	5.54 ***	39.0
Broccoli	1.85	0.76	40	0.91	0.18	13	0.669, 1.194	7.13 ***	49.1
Lettuce	1.35	0.44	40	1.35	0.13	14	0.242, 0.554	5.13 ***	51.4
Corn	0.63	0.27	40	0.28	0.04	8	0.257, 0.440	7.67 ***	45.2
Celery	1.35	0.49	40	0.99	0.0	11	0.206, 0.522	4.67 ***	39.0
Cucumber	0.67	0.29	40	0.66	0.24	14	−0.175, 0.171	−0.02	52.0
Cabbage	1.38	0.71	39	0.99	0.0	14	0.155, 0.618	3.38 **	51.0
Cauliflower	2.09	0.76	40	0.99	0.0	9	0.857, 1.341	9.18 ***	47.0
Whole-Milk	0.03	0.01	40	0.02	0	14	0.007, 0.113	9.05 ***	52.0
LF 2% Milk	0.03	0.01	40	0.02	0	14	0.006, 0.010	8.43 ***	52.0
100% Juice	0.05	0.01	40	0.03	0	13	0.020, 0.27	12.43 ***	39.0
Juice Drink	0.03	0.01	39	0.02	0.0	11	0.012, 0.016	13.29 ***	39.0
Reg. Ground Beef	0.26	0.04	40	0.23	0.05	7	−0.005, 0.060	1.71	45.0
Reg. Hot-Dog	0.26	0.12	40	0.08	0	14	0.146, 0.223	9.74 ***	39.0
LF Hot-Dog	0.28	0.12	40	0.08	0	14	0.158, 0.236	9.92 ***	39.0
Reg. Frozen Dinner	0.26	0.06	40	0.06	0.00	14	0.176, 0.214	20.43 ***	40.2
LF Frozen Dinner	0.23	0.05	36	0.06	0.00	14	0.155, 0.189	20.54 ***	36.4
Bagel	0.55	0.16	39	0.14	0.04	12	0.362, 0.470	15.40 ***	47.5
Danish	0.92	0.26	40	0.13	0.01	13	0.708, 0.875	19.13 ***	39.3
100% WW Bread	0.16	0.03	40	0.25	0	13	−0.097, −0.785	−19.16 ***	39.0
White Bread	0.15	0.02	40	0.20	0.01	14	−0.560, −0.389	−11.10 ***	51.2
Reg. Coke Soda	0.04	0.01	40	0.02	0.00	13	0.137, 0.172	17.75 ***	39.0
Diet Coke Soda	0.04	0.01	40	0.02	0.00	14	0.137, 0.172	17.75 ***	39.0
Reg. Cereal	0.38	0.38	40	0.17	0.10	14	0.006, 0.423	2.06 *	52.0
Low-Sugar Cereal	0.36	0.45	40	0.20	0.22	14	−0.938, 0.415	1.27	52.0
Reg. Chips	0.36	0.06	40	0.30	0.15	14	−0.032, 0.144	1.36 *	52.0
Baked Chips	0.54	0.41	33	0.09	0.03	14	0.229, 0.673	4.09 ***	45.0

M = mean; SD = standard deviation; df = degrees of freedom; Reg. = Regular; LF = Low fat; WW = whole wheat. ***** = *p* < 0.05, ** = *p* < 0.01, *** = *p* < 0.001.

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
