# Peer review of "Healthy Food Options at Dollar Discount Stores Are Equivalent in Quality and Lower in Price Compared to Grocery Stores: An Examination in Las Vegas, NV"

_ijerph, 2018, doi:10.3390/ijerph15122773_

Reviewer 1 Report

Overall, this paper addresses an important topic (existing landscape of food access in low income communities) and identifies potential solutions within communities that  may help alleviate the problem of low access to nutritious foods. The data collected are robust and significant, from a representative geography of the metro area of study (Las Vegas), and using a replicable methodology. The results and discussion are clearly presented, and implications of the research are well stated. 

More specific comments below: 

- Not clear in Abstract how these findings are "increasing" access to healthy foods, when it seems the study is just investigating what resources are already in place in the community. To increase access, would need to change something, i.e. make more produce available in dollar discount stores than there currently is. Consider revising language here. 

- Text/grammar in Abstract: opening sentence in Abstract should read: "access to and affordability of healthy foods."  

- Line 31- should be more specific: existing literature indicates access to affordable nutritious food is lacking in certain communities, typically designated "food deserts," not just "lacking" in general. 

- Lines 54-57- wording should be revised in this paragraph; second two sentences in particular are vague and lack contribution to the main point. Could consolidate by removing second sentence entirely. 

- Figure 1: could be labeled/titled to indicated that it is the Las Vegas Metro Area. 

- In the Conclusion, could discuss the implications of considering dollar discount stores as bigger part of community food resources a bit further; i.e. not considering a neighborhood to be a "food desert" because it has a dollar store might not be a good thing, as it might detract from funding or other efforts to make improvements in the food landscape. Could elaborate that one way to strengthen community food systems would be to further integrate dollar discount stores into food improvement efforts by adding nutrition labeling, eliminating stigma associated with shopping and dollar stores, or integrating dollar stores with urban ag scene to offer more locally produced foods from urban farms willing to partner/distribute foods there. 

Author Response

Reviewer 1

Overall, this paper addresses an important topic (existing landscape of food access in low income communities) and identifies potential solutions within communities that  may help alleviate the problem of low access to nutritious foods. The data collected are robust and significant, from a representative geography of the metro area of study (Las Vegas), and using a replicable methodology. The results and discussion are clearly presented, and implications of the research are well stated. 

More specific comments below: 

- Not clear in Abstract how these findings are "increasing" access to healthy foods, when it seems the study is just investigating what resources are already in place in the community. To increase access, would need to change something, i.e. make more produce available in dollar discount stores than there currently is. Consider revising language here. 

Thank you for the suggestion. We have revised the language to read “Findings indicate that dollar discount stores may be an existing community asset, and considering them as such may reveal that communities have increased access to healthy, affordable food.”

- Text/grammar in Abstract: opening sentence in Abstract should read: "access to and affordability of healthy foods."  

This has been corrected

- Line 31- should be more specific: existing literature indicates access to affordable nutritious food is lacking in certain communities, typically designated "food deserts," not just "lacking" in general. 

Thank you for pointing this out. We have corrected this.

- Lines 54-57- wording should be revised in this paragraph; second two sentences in particular are vague and lack contribution to the main point. Could consolidate by removing second sentence entirely. 

The second and third sentence have been modified and combined for clarity

- Figure 1: could be labeled/titled to indicated that it is the Las Vegas Metro Area. 

This has been added

- In the Conclusion, could discuss the implications of considering dollar discount stores as bigger part of community food resources a bit further; i.e. not considering a neighborhood to be a "food desert" because it has a dollar store might not be a good thing, as it might detract from funding or other efforts to make improvements in the food landscape. Could elaborate that one way to strengthen community food systems would be to further integrate dollar discount stores into food improvement efforts by adding nutrition labeling, eliminating stigma associated with shopping and dollar stores, or integrating dollar stores with urban ag scene to offer more locally produced foods from urban farms willing to partner/distribute foods there. 

This is a great recommendation. We have added it and it now reads “It is important for practitioners to understand the local food environment completely in order to document all opportunities for healthy food options, and to consider these non-traditional food outlets when assessing community food environments, designing and implementing nutritional interventions, and providing information or nutritional advice to those who do not live near traditional grocery stores. Additionally, considering dollar discount stores in food improvement efforts would strengthen the overall food system, such as nutrition labeling efforts or incorporation of urban agriculture efforts to offer more locally produced foods.

Reviewer 2 Report

This paper is clearly written and comprehensive. The topic is of public health interest and may encourage readers to think more broadly about accessibility to healthy food. The methodology is robust due to the use of the NEMS-S tool.

Some improvements to the understanding of the paper are:

Availability: Are there some foods which are less likely to be available at discount stores? Was there a reasonable variety of fresh fruit and vegetable items available at discount stores?

Price: State which foods have healthier and regular alternatives. I assume that it is all the non fruit and vegetable items except for ground beef. Were both  items of the pair generally available in the discount stores?

Line 98 - missing on (based on the acceptability)

Line 181, 182: Reference the statement that smaller neighbourhood markets have higher prices.

Line 198-205: Did you investigate whether the products are different brands were nutritionally similar? Or can you provide references for this statement. Line 200 has 'that' repeated.

References: Carefully check reference list as some have mistakes or are missing details (2, 14, 26, 32. 36.)

Author Response

Reviewer 2

This paper is clearly written and comprehensive. The topic is of public health interest and may encourage readers to think more broadly about accessibility to healthy food. The methodology is robust due to the use of the NEMS-S tool.

Some improvements to the understanding of the paper are:

Availability: Are there some foods which are less likely to be available at discount stores? Was there a reasonable variety of fresh fruit and vegetable items available at discount stores?

Thank you for your interest. While we did not do statistical analysis for each item, there were some things of note. We’ve added this to the paper. It now reads “Of note, no dollar discount stores carried pears and low fat ground beef, and 57% did not carry regular ground beef.”

Price: State which foods have healthier and regular alternatives. I assume that it is all the non fruit and vegetable items except for ground beef. Were both  items of the pair generally available in the discount stores?

Again, we did not do statistical analysis here, but aside from regular and low fat ground beef (now added to the availability section), both options were generally available. Additionally, all categories outside of produce have a healthy alternative. This is part of the NEMS tool. For example, low fat and full fat milk, low fat and full fat hotdogs, low sugar and regular sugar cereal.

Line 98 - missing on (based on the acceptability)

Thank you for catching this, it is fixed.

Line 181, 182: Reference the statement that smaller neighbourhood markets have higher prices.

A reference has been added.

Line 198-205: Did you investigate whether the products are different brands were nutritionally similar? Or can you provide references for this statement.

References have been added

Line 200 has 'that' repeated.

removed

References: Carefully check reference list as some have mistakes or are missing details (2, 14, 26, 32. 36.

These have been corrected.

Reviewer 3 Report

Please see the comments attached.

Author Response

Reviewer 3

Main comments It is a very interesting paper presenting original and new results, that fit well to the purpose of the journal. The conclusions are highly relevant to tackle social inequalities in nutrition and health, but would warrant extended discussion with reference with similar studies conducted in other location/other countries.

Conclusion 1) “While dollar-discount stores did have lower availability, they provided quality fresh and healthy foods which were usually less expensive. “ => “Dollar Discount Stores” seem to be very specific to the US, but it could be useful to readers from other countries to make some analogy with what exist in their countries. In particular I am thinking in hard discount stores in Europe (Aldi, Lidl, Dia.. ?), and in the selling of discount foods in regular supermarket chains in many countries. For instance, some studies found that prices differences between branded foods and generic or lowcost foods are not associated with parallel differences in nutritional quality. Another study found that diets aren’t healthier when they contain branded foods. Hence, compared with branded products, non-branded alternatives present better nutritional quality for their price and should therefore be preferred options to improve dietary quality when the budget for food is low.

See - Cooper S, Nelson M. ‘Economy’ line foods from four supermarkets and brand name equivalents: a comparison of their nutrient contents and costs. J Hum Nutr Diet. 2003;16:339–347. - Darmon N, Caillavet F, Joly C, et al. Low-cost foods: how do they compare with their brand name equivalents? A French study. Public Health Nutr 2009;12: 808–815. - Waterlander WE, van Kouwen M, Steenhuis IHM. Are diets healthier when they contain branded foods? British Food Journal · October 2014

Thank you for this recommendation. It has been added to the discussion section. It now reads “While dollar discount stores, pricing all items at a fixed $1 price, may be more common in the US, findings are still pertinent to other countries. For example, non-fixed price discount stores, as well as full service grocery stores that sell non-brand name food items are likely to offer equally healthy foods at a lower price. Hence, compared with branded products, non-branded alternatives are nutritionally equivalent options for a better price, and should therefore be preferred options to improve dietary quality when the budget for food is low.”

Conclusion 2) practitioners should consider non-traditional outlets when assessing the community food environment and designing and implementing outreach programs, as they may bridge some disparities in access. => This conclusion is important, but it may imply that nothing similar has been done previously. I think that the authors have to check. I know at least one successful intervention study conducted in discount stores which combined shelf labeling with a social marketing strategy to promote inexpensive foods with good nutritional quality (Gamburzew A, Darcel N, Gazan R, Dubois C, Maillot M, Tomé D, Raffin S, Darmon N. In-store marketing of inexpensive foods with good nutritional quality in disadvantaged neighborhoods: increased awareness, understanding and purchasing. Int J Behav Nutr Phys Activity, 2016, 13:104.), and I there are also very inspiring in-store interventions in poor populations conducted by John Gittelsohn, and colleagues.

This is very true, thank you for pointing it out. There have been studies examining non-traditional outlets, so we have reworded to simply state dollar discount stores. It now states “Practitioners should consider dollar discount stores when assessing the community food environment and designing and implementing outreach programs, as they may bridge some disparities in access.”

2 Specific comments Title I would remove quality from the title, because it is only “perceived quality” and it was only studied for produces (if I understand correctly). Alternatively, you could use the term ‘perceived quality”

The NEMS protocol does provide instruction on how to determine if the produce is acceptable or not (used for quality score). The definitions are below. The authors do not feel that this is necessarily perceived quality.

Acceptable = peak condition, top quality, good color, fresh, firm and clean ™ Unacceptable = bruised, old looking, mushy, dry, overripe, dark sunken spots in irregular patches or cracked or broken surfaces, signs of shriveling, mold or excessive softening

Abstract The abstract is difficult to read because some definitions are lacking: - Provide a simple and short explanation of what is a “dollar-discount store”. Line 4, you state that they are “stores that sold produce at fixed $1 price”: is this the definition?

Yes, this is the definition. We have moved it to the second sentence to improve clarity.

- Explain what is the NEMS-S, and that it is made of 3 “sub-scores” assessing respectively availability, quality and price, leading to a “total score” when the sub-scores are summed (isn’t it?). And then use this terminology throughout the paper.

The explanation of the subscales and summary score have been added. Similar language is used throughout, though we added price per item (not on NEMS), so that terminology is also used.

- Explain what is “M”

Done

Introduction - Line 67: ref 16 has to be introduced here.

done

Methods - Lines 93-94: introduce the terminology of sub-scores. It will be easier to read.

done

- Line 109: the statement that “all items were recalculated to price per piece” is not always true: it is for fruit but not for bread (lines112-113).

We have added that perishable and non-perishable items were recalculated to price per ounce

Discussion - Line 181: replace have higher prices by are likely to have higher prices, or include a reference.

We added a reference

- Line 193: price modification is not the only tool, and it is not an easy tool to implement for the long term. Social marketing of foods that already are healthy and inexpensive is another option, probably easier to implement because it doesn’t imply additional costs, neither for the shopper, nor for the vendor, nor for the public authorities (Gamburzew A, Int J Behav Nutr Phys Activity, 2016, 13:104.).

Discussion of this tool and suggested reference have been added

- Line 201-202: it seems that references 31-34 apply to the first part of the sentence (“consumers perceive brand names to be of higher quality”) but additional references should be added to document the second part of the sentence (“even when they aren’t)=> For instance, Cooper S et al J Hum Nutr Diet. 2003;16:339–347; - Darmon N, et al Public Health Nutr 2009;12: 808–815; Waterlander WE et al 2014 British Food Journal)

These references have been added

- One important thing is lacking in the discussion: the fact that the total score of NEMS-S is not relevant when one wants to evaluate the capacity of a store to provide good nutrition at good price for low-income people. The sub-scores seem to be more informative.

Thank you for pointing this out. A brief discussion of this has been added
